# A knowledge graph construction method for compliance review of water conservancy project reports

Zelin Ding[1], Zhefei Fan[1]*, Yuanfeng Hao[2], Tao Wang[1‡], Xin Du[1‡], Xinhang Zhang[1‡]

1 North China University of Water Resources and Electric Power, Zhengzhou City, Henan Province, China, 2 Henan Water & Power Engineering Consulting CO., Ltd, Zhengzhou City, China

☯ These authors contributed equally to this work.
‡ TW, XD and XZ also contributed equally to this work.
* 2450365632@qq.com

## Abstract

To break through the efficiency and accuracy bottlenecks of manual mode in the compliance review of water conservancy project reports and promote the digital transformation of "Smart Water Conservancy", this paper proposes a knowledge graph construction method for the compliance review of water conservancy project reports. Firstly, based on natural language processing technology, the BERT-BiLSTM-CRF model is used for named entity recognition to accurately locate key entities such as engineering parameters and normative clauses. Secondly, the context-free grammar (CFG) is used to parse the logical relationships between entities, and the normative clauses are transformed into "head entity + relationship + tail entity" triples through a semantic label system to achieve structured expression of knowledge in the water conservancy field. Finally, the Neo4j graph database is used to store the knowledge graph, and the Py2neo toolkit is used to complete the efficient import and dynamic update of triple data. The research takes the actual review of water conservancy project reports as a case to verify the feasibility of the method. Practice has proved that this method effectively improves the efficiency and accuracy of the compliance review of water conservancy project reports, providing technical support and practical reference for the digital transformation of water conservancy projects, and is of great significance for promoting the intelligent development of the water conservancy industry.

## Introduction

Against the backdrop of the accelerated promotion of the "smart water conservancy" strategy and the national digital transformation, the compliance review of water conservancy engineering reports, as the core link of engineering construction quality control, still relies on the traditional mode of manually comparing normative

**Data availability statement:** ll relevant data are within the manuscript and its Supporting Information files.

**Funding:** Postgraduate Education Reform and Quality Improvement Project of Henan Province (YJS2026YBGZZ03).

**Competing interests:** The authors have declared that no competing interests exist.

provisions word by word, facing prominent problems such as low efficiency, strong subjectivity, and difficulty in accurately capturing complex logical relationships [1]. With the expansion of water conservancy projects and the complexity of regulatory systems, traditional review models are no longer able to meet the needs of intelligent and precise management, and there is an urgent need to introduce artificial intelligence technology to break through bottlenecks [2].

In recent years, with the rapid development of artificial intelligence in the field of natural language, technologies such as knowledge graphs for processing unstructured correlated data have been widely applied [3]. In 2012, Google released a large-scale knowledge graph, which sparked a craze for knowledge graph research and application. Various types of knowledge graphs such as DBpedia, YAGO, NELL, ArnetMiner, etc. emerged one after another [4]. With the advancement of research, knowledge graphs in the era of deep learning attempt to introduce neural networks, such as Bordes et al.'s knowledge base structure embedding and BORDES A et al.'s Neural Tensor Network (NTN), which embed entities and relationship words for node representation and triplet judgment, achieving significant results in knowledge graph graph graph completion tasks [5]. In recent years, thanks to advances in natural language processing, pre trained models such as BERT have improved their text understanding and retrieval capabilities, making it possible to understand and reason on raw text. For example, CHEN D et al.'s DrQA directly extracts question answers from text [6]. In the field of hydraulic engineering, research on knowledge graph construction is constantly deepening. Although existing problems have been identified and research frameworks have been proposed, there is relatively little direct application of knowledge graph technology in the compliance review of hydraulic engineering reports. Knowledge graph, as a semantic modeling technology based on graph structure, integrates fragmented normative knowledge into a structured network through the "entity relationship attribute" triple system, providing a new path for solving the above problems.

Therefore, this article first preprocesses the report data, and then uses natural language processing techniques to identify entities through the BERT BiLSTM CRF model. Combined with context free parsing, the article proposes seven semantic role labels to organize the logical structure of the text. Simultaneously convert the rules into a knowledge graph triplet, and finally construct a review knowledge graph using Neo4j and Py2neo. This research innovates the knowledge extraction and application mode, realizes intelligent retrieval of standardized clauses and automatic comparison of drawings, effectively improves review efficiency, and promotes the intelligent development of water conservancy engineering [7].

## Research method

### Semantic syntactic joint parsing method

In the field of deep learning, the BERT BiLSTM CRF model is a widely used architecture in natural language processing, performing well in tasks such as text classification and named entity recognition. Its network structure is divided into three modules:

BERT, Bidirectional Long Short Term Memory Network (BiLSTM), and Conditional Random Field (CRF). The working principle is to extract word vectors as basic features through a BERT pre trained model, input them into BiLSTM for deep feature extraction, and then decode and predict the optimal annotation sequence through the CRF module [8]. Example of Sample Division for Hydraulic Standards, as shown in Fig 1.

(1) BERT adopts a Transformer architecture based on "Self Attention", and its encoder achieves deep representation learning of input sequences through self attention layers. In the calculation process, the embedding vectors of each word element are mapped to the query matrix Q, key matrix K, and value matrix V, and the correlation weights between word elements are calculated using the attention function shown in formula (1). In order to prevent the phenomenon of numerical instability in attention scores as the embedding dimension dk increases, a dimension scaling factor is introduced in the calculation process. Subsequently, the attention weights are normalized using the Softmax function and weighted with the value matrix to obtain a lexical representation that integrates global contextual information.

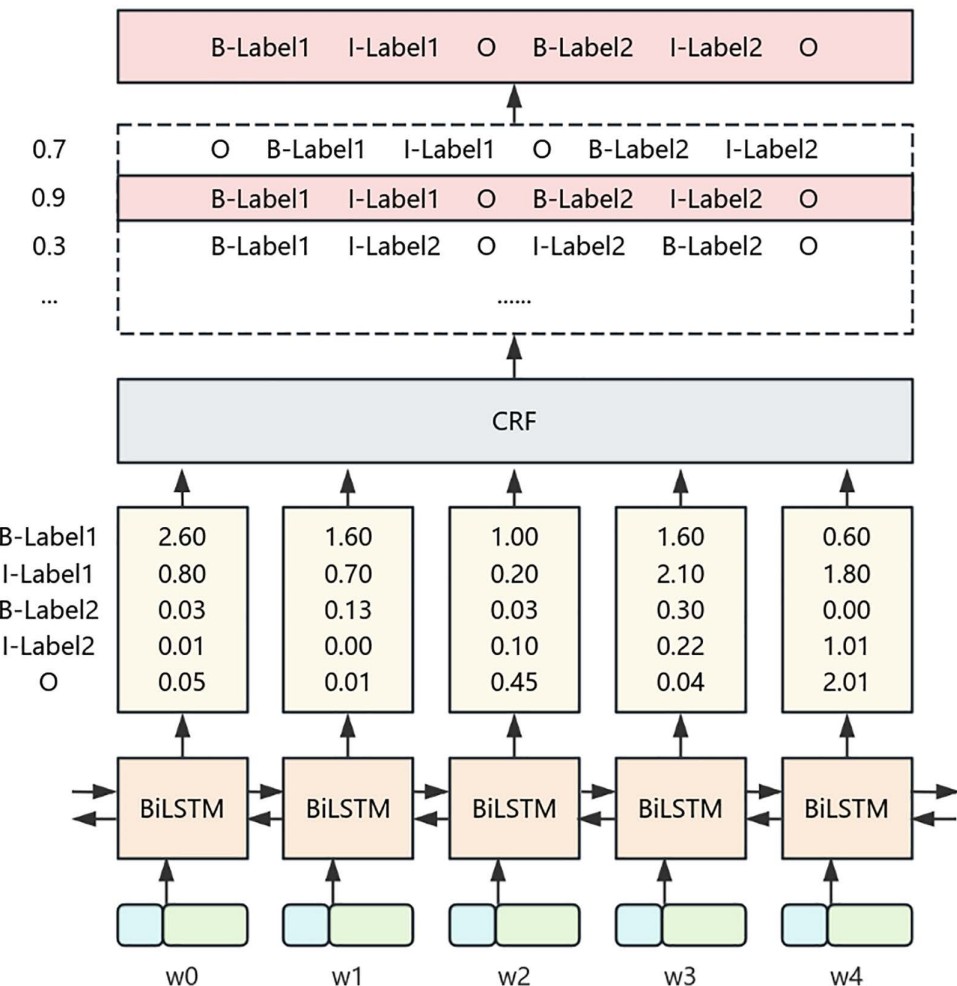

**Fig 1. Example of Sample Division for Water Conservancy Standards.**

$$Attention(Q, K, V) = Softmax\left(\frac{QK^T}{\sqrt{d_k}}\right)V \tag{1}$$

BERT further proposed the Multi attention mechanism by introducing self attention mechanism. The calculation formula is as follows.

$$MultHead(Q, K, V) = Concat(head_1, \cdots, head_n)W^O \tag{2}$$

$$head_i = Attenton(QW_i^Q, KW_i^K, VW_i^V) \tag{3}$$

(2) LSTM (Long Short Term Memory Network) is an improvement on traditional RNN, which solves the problem of gradient explosion during RNN training, uses gating to achieve long-term memory, and avoids forgetting important features. It introduces three gating mechanisms for each hidden layer neuron based on RNN, selectively remembering and forgetting feature information. However, LSTM can only capture unidirectional sequence dependencies. Therefore, researchers propose BiLSTM, which consists of two independent LSTM layers. Forward and backward LSTMs process sequences in positive and reverse order, respectively, to capture contextual information from different directions.

$$f_t = \sigma(w_t[h_{t-1}, x_t] + b_f) \tag{4}$$

$$i_t = \sigma(w_i[h_{t-1}, x_t] + b_i) \tag{5}$$

$$\hat{C}_t = tanh(w_C[h_{t-1}, x_t] + b_C) \tag{6}$$

$$C_t = f_t * C_{t-1} + i_t * \hat{C}_t \tag{7}$$

$$O_t = \sigma(w_o[h_{t-1}, x_t] + b_o) \tag{8}$$

$$h_t = O_t tanh(C_t) \tag{9}$$

(3) In sequence annotation tasks, although BiLSTM can capture long-distance text dependencies, it independently predicts word element labels and cannot explicitly model label transfer constraints. The Conditional Random Field (CRF) introduces a label transition probability matrix and utilizes the dependency relationship between adjacent labels to output the globally optimal annotation sequence, which compensates for the shortcomings of BiLSTM. After receiving the output score of BiLSTM, CRF generates the maximum possible prediction sequence that meets the annotation transfer constraint. For the sequence X=(x1, x2..., xn), if the output score matrix P of BiLSTM is n × k (n is the number of words, k is the number of labels), $P_{ij}$ represents the score of the i-th word and j-th label, corresponding to the predicted sequence Y=(y1, y2,..., yn), the score function is:

$$Score(x, y) = \sum_{i=0}^{n} A_{yi,yi+1} + \sum_{i=1}^{n} P_{i,yi} \tag{10}$$

$A_{i,j}$A is the transition score matrix, which is the score for the transition from label i to label j, and the value of A is k+2. The probability of generating the predicted sequence Y is:

$$p(Y|X) = \frac{e^{s(x,y)}}{\sum_{\widetilde{Y} \in Yx} s(X, \widetilde{Y})}$$

(11)

Take the logarithm of both sides of equation 1.11 to obtain the likelihood function of the predicted sequence:

$$In(p(Y|X)) = score(X, Y) - In(\sum_{\widetilde{Y} \in Yx} s(X, \widetilde{Y}))$$

(12)

In the formula, $\widetilde{Y}$ represents the real annotation sequence, and YX represents all possible annotation sequences. The output sequence with the highest score obtained after decoding:

$$Y^* = \underbrace{argmax}_{\widetilde{Y} \in Yx} s(X, \widetilde{Y})$$

(13)

After completing dynamic semantic annotation and entity extraction, the text has been transformed into a sequence of entities with clear semantic labels, but the logical relationships between entities are still hidden in the sequence structure and have not yet formed a structured logical expression. To further reveal the hierarchical relationships and semantic constraints between entities, it is necessary to construct a syntax tree to transform linear text into a tree like logical structure, so that the multi-level relationship of "object attribute constraint" can be explicitly presented. Context free grammar (CFG) is a formal language description method that uses formal rules to define valid strings. Its production rule consists of a single non terminal character on the left, ensuring the context independence of syntax inference. CFG is widely used in fields such as natural language processing [9]. CFG is defined as a quadruple G=(V, T, P, S), where V and T are the sets of non terminal symbols and terminal symbols respectively, P is the set of production equations, and S is the initial symbol. Non terminal symbols represent abstract components, while terminal symbols are concrete instances. Production equations implement the derivation of non terminal symbols from strings, and syntax trees can visually display the derivation process. They consist of roots (starting symbols), interiors (non terminating symbols), and leaf nodes (terminating symbols). When designing syntax, priority should be clearly defined, and hierarchical grouping can reflect priority relationships.The example of the CFG (Context-Free Grammar) syntax tree is as shown in Fig 2.

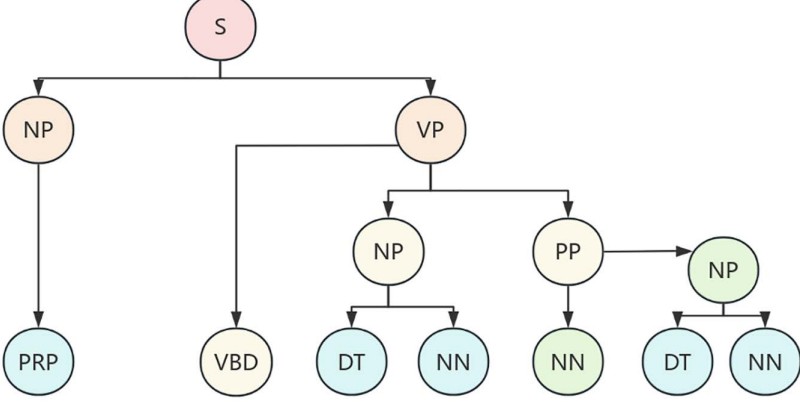

**Fig 2. Example of Sample Division for Water Conservancy Standards.**

## Rule extraction integration

The language used to describe standard articles mainly includes conditional sentences and compound sentences. Conditional sentences contain both simple and compound forms, with compound sentences consisting of multiple simple sentences, and master-slave compound sentences being more common. These sentence structures all contain four elements: the inspected object, attributes, comparison operation constraints, and requirement limits. They can be marked with semantic tags obj, prop, cmp, and rprop, and parsed into a structure of "the attribute (prop) of the object (obj) should satisfy the requirement (rprop) through the relationship (cmp)" and represented in a tree.

There are two ways to extend complex specification statements:

(1) Compound Structure.For example, "The bottom elevation of the beam-slab of the service bridge should be 0.5m higher than the highest flood level". At this time, a tag sobj can be added to indicate that "beam-slab" is the upper-level object of "obj". Meanwhile, sobj can be reused, where the sobj on the right is the sub-object of the sobj on the left, as shown in Fig 3 (a). In addition, rprop may also be a lower-level attribute of an object. For instance, "The top elevation of the sluice should not be lower than the top elevation of the flood dike". Here, "top elevation" indicating the required limit value is an attribute of "flood dike". Therefore, a tag robj can be added to mark "flood dike" as the upper-level object of rprop, as shown in Fig 3 (b).

(2) Conditional Structure.As shown in Fig 3 (c), in the sentence "If the loess foundation has no filter layer, the seepage path coefficient shall not be less than 4", "has no filter layer" does not indicate a required limit value. Instead, it means the subsequent review will be implemented only when this condition is met, similar to the "If" part in an If-Then structure. Therefore, a tag arprop can be added to mark such elements representing "If"-type preconditions. Meanwhile, like rprop, robj can also be used to mark the upper-level object of arprop.

In summary, this article proposes seven semantic tags to represent the different semantic roles of words (or phrases) in normative provisions. Among them, the obj, sobj, and prop elements are used to represent the elements to be checked in the drawing. obj can be the child node of sobj and the parent node of prop. CMP is the comparison or existence relationship between the element prop to be checked and the required conditions Rprop/Rprop. Rprop and ARprop respectively represent two types of requirement conditions. Rprop is a constraint requirement applied to prop, while ARprop is a prerequisite applied to prop. The explanations of each semantic tag are shown in Table 1.

Each semantic tag can represent an entity, and the relationship between entities can be defined to form a triple structure of "head entity+relationship+tail entity". It should be noted that cmp itself has the meaning of "relationship", so it can be used as the relationship between prop and rprop/arprop to form the "prop+cmp+rprop" structure. Furthermore, based on the meanings of other semantic tags, three other basic relationships can be defined, namely inclusion, existence, and condition. The basic relationship types and examples are shown in Table 2.

**Fig 3. Example of Semantic Annotation.**

**Table 1. Semantic Label Explanation.**

| tag name | Label State |
|---|---|
| sobj | Sub level pending review objects, mainly reviewing entities |
| obj | Parent level pending object, the upper level of a child level object, usually serves as a modifier |
| prop | Pending attribute, indicating a certain characteristic possessed by the pending object |
| cmp | Arithmetic relationship, a term used for comparing quantitative values |
| rprop | Sub level constraint requirements, mainly checking the content |
| robj | Parent level constraint requirements, the upper level of child level constraints, usually serve as a modifier |
| arprop | When the condition for conducting the review is true, proceed to the next step of the review |

**Table 2. Basic Relationship Example.**

| Head entity | Tail entity | relationship type | Example |
|---|---|---|---|
| sobj | obj | contain | <Work Bridge, including beams and slabs> |
| robj | rprop/arprop | exist | <Flood control embankment, existing, top elevation> |
| obj | prop | exist | <Water gate, existing, top elevation> |
| arprop | Prop/obj | condition | <Water blocking, conditions, normal storage level> |
| prop | rprop/arprop | <cmp> | The bottom elevation should be higher than the maximum flood level by 0.5m |

In summary, the following methods have been developed to extract review rules from normative provisions: firstly, semantic annotation of normative provisions is performed using a trained BERT-BILSTM-CRF named entity recognition model; Next, the CFG analysis method is used to parse the annotated normative clauses, and the parsing results are represented using a tree structure. The purpose of this step is to achieve formal expression of normative clause sentences; Finally, the tree structured statements are mapped to the corresponding nodes and relationships in the knowledge graph through matching, as shown in Fig 4.

## Recognition model training

### Data preprocessing

Select the specifications described in Table 3 as the training data for the semantic annotation model. Before training, it is necessary to preprocess the specification file into text data [10,11], which mainly includes the following processes:

(1) Remove content unrelated to the article and only retain the section containing the review rules. For example, deleting contents such as table of contents, preface, and numbering from the specifications.

(2) For some review articles presented in table form, the content of the table should be organized and converted into corresponding simple sentences. For example, in Table 3, it can be converted into rule texts with simple sentence structures such as "The safe increase value of the normal water level when the third level water gate blocks water is not less than 0.4m" and "The safe increase value of the highest water level when the fourth and fifth level water gates block water is not less than 0.2m".

(3) Break down the standard clauses of compound sentences into simple clause clauses. For example, in the article "The minimum thickness of concrete or reinforced concrete pavement should be greater than 0.4m, and the permanent seam spacing along the water direction can be used between 8-20m", it can be divided into the article "The minimum

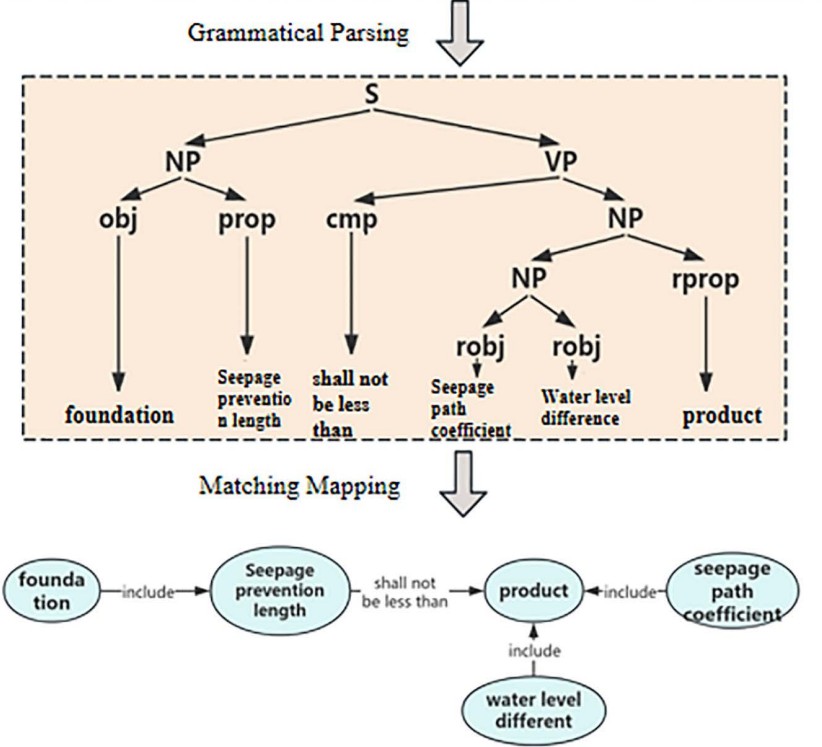

**Fig 4. Schematic diagram of knowledge extraction method for review.**

**Table 3. High and Low Limits of Safety Prices for Water Gates (Unit: m).**

| Operation | | Water gate level | | | |
|---|---|---|---|---|---|
| | | Level 1 | level 2 | Level 3 | Level 4, Level 5 |
| When blocking water | normal storage level | 0.7 | 0.5 | 0.4 | 0.3 |
| | Maximum storage level | 0.5 | 0.4 | 0.3 | 0.2 |
| When releasing water | design flood level | 1.5 | 1.0 | 0.7 | 0.5 |
| | Check flood level | 1.0 | 0.7 | 0.5 | 0.4 |

thickness of concrete or reinforced concrete pavement should be greater than 0.4m" and the article "The permanent seam spacing along the water direction of concrete or reinforced concrete pavement can be used between 8-20m", which can then be further divided into the article "The minimum thickness of concrete pavement should be greater than 0.4m", the article "The minimum thickness of reinforced concrete pavement should be greater than 0.4m", and

the article "The permanent seam spacing along the water direction of pavement can be used between 8-20m". When preprocessing normative texts, the simpler the rule provisions are split, the easier it is to extract knowledge from normative provisions in the future

## Semantic annotation

Semantic annotation refers to the process of assigning semantic labels manually or automatically to preprocessed water conservancy engineering specification data, clarifying semantic information such as entities, attributes, relationships, and logical conditions in the text. Named entity recognition methods based on deep learning can essentially be classified as a sequence labeling problem. Sequence labeling assigns labels to each element in a given text sequence based on its semantic and grammatical features, aiming to construct the corpus required for model training. Through the analysis of the seven types of semantic labels defined in the previous sections of the sluice engineering specifications, this study adopts the BIO annotation system (assuming X is an entity, B-X represents the beginning of the entity, I-X represents the middle or end of the entity, and O represents a non-entity) to perform semantic annotation on the specification provisions. Fig 5 shows an example of BIO annotation, and the text annotation example and corresponding JSON text are presented in Fig 6.

Randomly divide the 472 annotated statements into layers in an 8:2 ratio. Among them, 80% is the training set (Train) with a total of 656 statements, and 20% is the test set (Valid) with a total of 164 statements. The training set is used to train and update deep learning models, while the validation set is used to test model performance. It should be noted that when randomly dividing statements, the training/testing ratio of each label should be as close as possible to 8:2. Table 4 shows the number and proportion of each BIO semantic label in the randomly partitioned training and validation sets. It can be seen that the training/testing ratio of each label is close to 8:2, which basically meets the requirements of model training.

## Training results

Semantic annotation based on deep learning requires setting the learning rate and batch size. Learning rate is a key hyperparameter in machine learning that controls the step size of model parameter updates. It determines the magnitude of parameter adjustment of the model based on the gradient of the loss function in each iteration. A suitable learning rate can balance convergence speed and stability, which is an important foundation for efficient model training. Batch size refers to the number of training samples used each time the model parameters are updated. In practice, batch size and learning rate often need to be adjusted collaboratively to optimize training effectiveness and efficiency. For the task of semantic annotation of standardized knowledge, the BERT-BILSTM-CRF named entity recognition model is adopted, and the parameter settings of the model are shown in Table 5. Based on previous experimental studies [12–14], two hyperparameters, learning rate and batch size, were selected for ablation experiments. The evaluation criteria used in the experiment are the same as above, and it runs on a 64 bit Windows system and NVIDIA GeForce RTX 4090 memory. The hyperparameter settings of the experiment and the running results of the model on the validation set are shown in Table 6.

After testing, the model showed the best performance after 15 rounds of training (Epoch), with the optimal hyperparameter combination being "learning rate=3e-5" and "batch size=4". In the setting of this parameter combination, the semantic annotation results of the model on the validation set are shown in Table 7, and the corresponding confusion matrix is shown in Fig 7.

The length of each section on the rock foundation should not exceed 20 meters.
B-obj I-obj O O B-prop I-prop I-prop I-prop B-cmp I-cmp I-cmp I-cmp B-rprop I-rprop

**Fig 5. Article BIO annotation example.**

**Fig 6. Text annotation example and corresponding JSON text.**

**Table 4. Number and proportion of BIO semantic tags.**

| BIO semantic tags | total number | Number of training sets | Verification set quantity | Training/validation quantity ratio |
|---|---|---|---|---|
| B-sobj | 138 | 108 | 30 | 3.6 |
| I-sobj | 498 | 404 | 94 | 4.3 |
| B-obj | 723 | 584 | 139 | 4.2 |
| I-obj | 2166 | 1724 | 442 | 3.9 |
| B-prop | 915 | 732 | 183 | 4.0 |
| I-prop | 2317 | 1824 | 493 | 3.7 |
| B-cmp | 649 | 508 | 141 | 3.6 |
| I-cmp | 1661 | 1315 | 346 | 3.8 |
| B-rprop | 658 | 521 | 137 | 3.8 |
| I-rprop | 1688 | 1286 | 402 | 3.2 |
| B-robj | 124 | 97 | 27 | 3.6 |
| I-robj | 626 | 510 | 116 | 4.4 |
| B-arprop | 270 | 216 | 54 | 4.0 |
| I-arprop | 956 | 761 | 195 | 3.9 |
| O | 3973 | 3194 | 780 | 4.1 |

Based on the above results, the comprehensive accuracy, recall rate, and F1 of the model are 86.6%, 87.2%, and 86.8%, respectively. Among these tags, only cmp, Rprop, and robj achieved F1 scores of over 90%, while the F1 scores of other semantic tags were relatively low, ranging from 77.6% to 89.4%. This result indicates that deep learning based semantic annotation methods can be applied to large-scale semantic annotation of long and complex sentences, and can

**Table 5. Super parameter setting and operation results of Ablation Experiment.**

| control group | Learning rate | Batch size | F1-score |
|---|---|---|---|
| Setting 1 | 3e-5 | 2 | 0.845 |
| Setting 2 | 5e-5 | 2 | 0.862 |
| Setting 3 | 7e-5 | 2 | 0.850 |
| Setting 4 | 3e-5 | 4 | 0.868 |
| Setting 5 | 5e-5 | 4 | 0.824 |
| Setting 6 | 7e-5 | 4 | 0.858 |
| Setting 7 | 3e-5 | 8 | 0.833 |
| Setting 8 | 5e-5 | 8 | 0.832 |
| Setting 9 | 7e-5 | 8 | 0.864 |

**Table 6. Named entity recognition model parameter settings.**

| parameter | numerical value | meaning |
|---|---|---|
| LSTM-size | 128 | Dimension of LSTM hidden layer |
| epoch | 15 | number of iterations |
| max_seq_length | 128 | Maximum sample length |
| dropout_rate | 0.5 | Make the activation value of neurons fail with a probability of 0.5 |

**Table 7. Semantic annotation results of the model on the validation set.**

| Semantic tags | quantity | Precision | Recall | F1-score |
|---|---|---|---|---|
| sobj | 30 | 0.767 | 0.785 | 0.776 |
| obj | 139 | 0.795 | 0.913 | 0.850 |
| prop | 183 | 0.883 | 0.905 | 0.894 |
| cmp | 141 | 0.950 | 0.949 | 0.949 |
| rprop | 137 | 0.880 | 0.952 | 0.915 |
| robj | 27 | 0.934 | 0.928 | 0.931 |
| arprop | 54 | 0.845 | 0.787 | 0.815 |
| O | 780 | 0.874 | 0.759 | 0.813 |
| amount to | 3578 | 0.866 | 0.872 | 0.868 |

obtain relatively accurate results. In addition, the small dataset used in this study may not fully utilize the performance of deep learning. Overall, semantic annotation based on deep learning can currently achieve satisfactory results and has great potential for further improving performance.

## Model verification experiments and innovation demonstration

To clarify the innovation and superiority of the proposed BERT-BiLSTM-CRF model in the semantic annotation task of water conservancy specifications, this section designs baseline comparison experiments and ablation experiments based on the annotated dataset (656 samples in the training set and 164 samples in the test set) constructed in the previous sections and the determined optimal hyperparameters (learning rate = 3e-5, batch size = 4), verifying the innovation of the model from two dimensions: "horizontal model comparison" and "vertical module contribution".

### Basic experimental settings

**(1) Dataset and evaluation metrics.** This study adopts the annotated dataset and BIO annotation system from the previous sections, focusing on the recognition performance of 7 core semantic labels (sobj, obj, prop, cmp, rprop, robj,

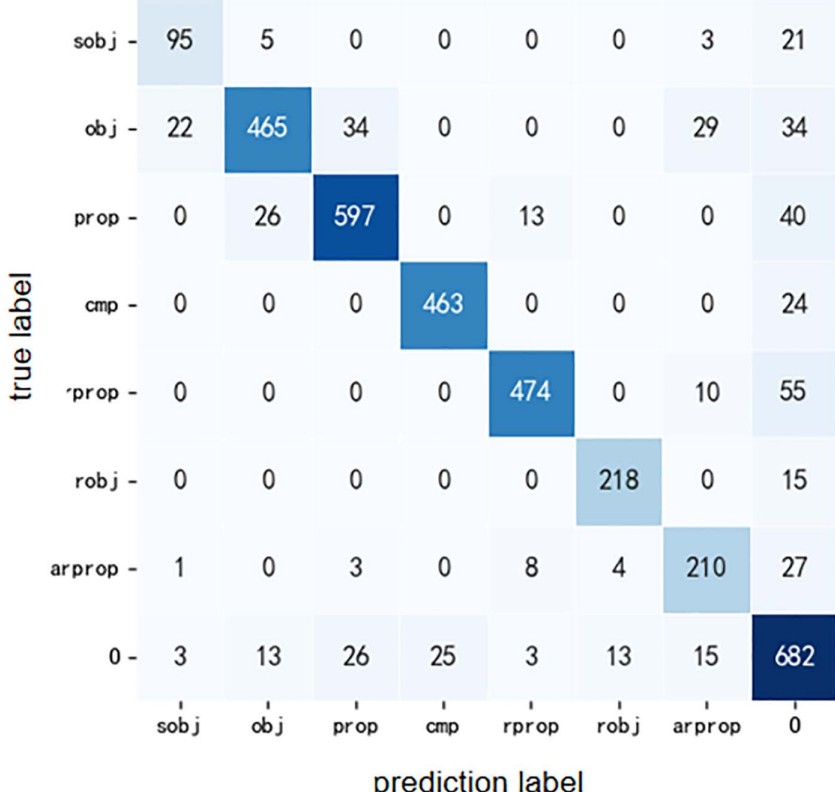

**Fig 7. Confusion matrix of semantic annotation results on verification set.**

arprop). The evaluation metrics include Precision, Recall, and F1-score, and additionally supplements the "Accuracy of Hydraulic Professional Entity Recognition" (for the recognition accuracy of domain-specific entities such as "sluice crest elevation" and "seepage path coefficient") to ensure that the evaluation dimensions meet the needs of water conservancy engineering scenarios.

**(2) Experimental environment and parameter benchmark.** The experimental environment is consistent with that in the previous sections (64-bit Windows system, NVIDIA GeForce RTX 4090 graphics card). The basic model parameters adopt the optimal configuration in Table 5 (LSTM-size = 128, epoch = 15, max_seq_length = 128, dropout_rate = 0.5), only adjusting the model structure to construct baseline models and ablation groups to ensure the fairness and comparability of the experiments.

## Baseline model comparison experiments

Three types of representative models are selected as baselines to verify the innovation of the proposed model's "BERT pre-training + BiLSTM bidirectional modeling + CRF label constraint" combination. Performance tests of the above baseline models are conducted on the test set, and the results are shown in the following Table 8:

The F1-score of the proposed model is 9.2 percentage points higher than that of BERT (Baseline 1). This indicates that the bidirectional gating mechanism of BiLSTM can effectively capture the long-distance dependencies between "entity-attribute-constraint" in water conservancy specifications (e.g., the association between "normal water storage level safety margin" and "0.4m"), while the label transition constraint of the CRF layer reduces the mislabeling rate of "arprop (prerequisite condition) and rprop (core constraint)" from 17.8% to 6.9%.

**Table 8. Performance Comparison between the Proposed Model and Baseline Models.**

| Model | Precision | Recall | F1-score | Accuracy of Hydraulic Professional Entity Recognition |
|---|---|---|---|---|
| BERT (Baseline 1) | 0.783 | 0.769 | 0.776 | 0.754 |
| BERT-LSTM (Baseline 2) | 0.817 | 0.805 | 0.811 | 0.795 |
| BiLSTM-CRF (Baseline 3) | 0.798 | 0.79 | 0.794 | 0.781 |
| Proposed BERT-BiLSTM-CRF | 0.866 | 0.872 | 0.868 | 0.849 |

The F1-score of the proposed model is 5.7 percentage points higher than that of BERT-LSTM (Baseline 2), confirming that BiLSTM is more suitable for the semantic logic of complex sentences in water conservancy specifications (e.g., "If there is no filter layer in the loam foundation, the seepage path coefficient should not be less than 4"). It can simultaneously cover the association between "preceding condition and subsequent constraint", and the recognition accuracy of the conditional label "arprop" is increased by 7.2 percentage points.

The F1-score of the proposed model is 7.4 percentage points higher than that of BiLSTM-CRF (Baseline 3), reflecting the advantage of BERT pre-trained semantic representation. The contextual information learned from massive texts can accurately recognize professional terms such as "seepage path coefficient" and "filter layer", avoiding the ambiguity of entity boundaries caused by randomly initialized word vectors (e.g., the mis-splitting rate of "traffic bridge beam bottom elevation" is reduced from 11.5% to 2.8%).

## Core module ablation experiments

To verify the necessity and collaborative value of the three core modules (BERT pre-training layer, BiLSTM bidirectional layer, and CRF layer) in the proposed model, three ablation groups are designed. The contribution of each module is quantified by "removing core modules and observing performance changes".

The results of the ablation experiments are shown in the following Table 9:

The F1-score of Ablation 1 (Without BERT) decreases by 7.7 percentage points, and the average accuracy of key labels decreases by 8.3 percentage points. This indicates that the BERT pre-training layer provides an accurate semantic foundation for the recognition of hydraulic professional terms, which is the core to solving the problem of "domain entity ambiguity".

The F1-score of Ablation 2 (Without BiLSTM) decreases by 5.7 percentage points, and the average accuracy of key labels decreases by 5.9 percentage points. This confirms that the bidirectional modeling of BiLSTM can effectively capture cross-phrase semantic dependencies in water conservancy specifications (e.g., the conditional association between "Level 3 sluice water retaining" and "safety margin").

The F1-score of Ablation 3 (Without CRF) decreases by 4.1 percentage points, and the average accuracy of key labels decreases by 4.7 percentage points. This reflects that the label transition constraint of the CRF layer ensures the logical consistency of semantic labels, reducing mislabeling between "rprop and robj" and "arprop and obj".

**Table 9. Results of Core Module Ablation Experiments for the Proposed Model.**

| Model Configuration | Precision | Recall | F1-score | Average Accuracy of Key Labels (arprop/rprop/prop) |
|---|---|---|---|---|
| Proposed Model (BERT-BiLSTM-CRF) | 0.866 | 0.872 | 0.868 | 0.885 |
| Ablation 1 (Without BERT) | 0.793 | 0.789 | 0.791 | 0.802 |
| Ablation 2 (Without BiLSTM) | 0.812 | 0.81 | 0.811 | 0.826 |
| Ablation 3 (Without CRF) | 0.824 | 0.831 | 0.827 | 0.838 |

The above results show that the three core modules of the proposed model are not simply superimposed, but are innovatively adapted to the characteristics of water conservancy specifications (many professional terms, many long sentences, and strong logical constraints). The three modules collaboratively achieve "accurate semantic representation, comprehensive sequence modeling, and logical label constraint", collectively supporting the model's performance to be superior to baseline models with single modules or simplified structures.

## Experimental conclusions and innovation summary

Through baseline comparison and ablation experiments, the innovations of the proposed BERT-BiLSTM-CRF model are clarified as follows:

Structural Innovation: A collaborative architecture of "pre-trained semantic representation + bidirectional sequence modeling + global label constraint" is proposed, which specifically solves three pain points in the semantic annotation of water conservancy specifications: ambiguity in professional entity recognition, insufficient capture of long-sentence dependencies, and confusion in label logic.

Performance Advantage: The model achieves a comprehensive F1-score of 0.868 on the test set, which is 5.7%–9.2% higher than that of traditional baseline models. Moreover, the accuracy of hydraulic professional entity recognition reaches 84.9%, which is suitable for the practical needs of water conservancy engineering scenarios.

Application Value: The high-precision annotation results of the model (e.g., accurate distinction of 7 types of semantic labels) provide high-quality data input for syntax parsing and triple conversion in "Review Rule Analysis", directly supporting the effectiveness of the construction of the compliance review knowledge graph.

## Review rule storage

### Analysis of review rules

Based on the semantic annotation model trained in the previous section, this study automates the semantic annotation of standard clauses for water gates, thereby achieving the extraction of various entities. To further analyze the semantic relationships between entities, context free grammar is used to parse normative clauses, and based on this, a domain knowledge representation model is constructed. After analysis, the articles can be divided into five types: relational constraints, attribute numerical constraints, attribute proportion constraints, attribute nesting constraints, and conditional constraints.

(1) The existence relationship constraint clauses reflect the existence relationship between the examination objects, and use affirmation or negation to constrain the relationship between the subject examination object and the object examination object. For example, "a cushion layer should be set up under the bottom protection", "under the bottom protection" and "cushion layer" are the subject and object respectively, "setting" indicates the relationship, and "should" determines the connection between the two. From the perspective of auditing, computers can understand that there should be a "cushion layer" under the "bottom protection", otherwise it is not compliant. The corresponding knowledge graph triplet transformation is shown in Fig 8.

(2) The attribute value constraint clauses mainly reflect the comparative relationship between the attribute values of the object under review, and constrain the relationship between the object under review and its attributes with affirmation or negation. For example, "the inner diameter of the drainage pipe is greater than 0.2m", "inner diameter" and "0.2m" are the objects for review, "drainage pipe" limits the range of "well pipe", "inner diameter" is the attribute of "well pipe", and "greater than" indicates comparison. When the value of "inner diameter" is greater than "0.2m", it meets the requirements; otherwise, it does not comply. The knowledge graph triplet transformation is shown in Fig 9.

(3) Attribute proportion constraint clauses reflect the proportional relationship of review attributes to constrain pending attributes. For example, the length of the straight section of the river should not be less than 5 times the width of the

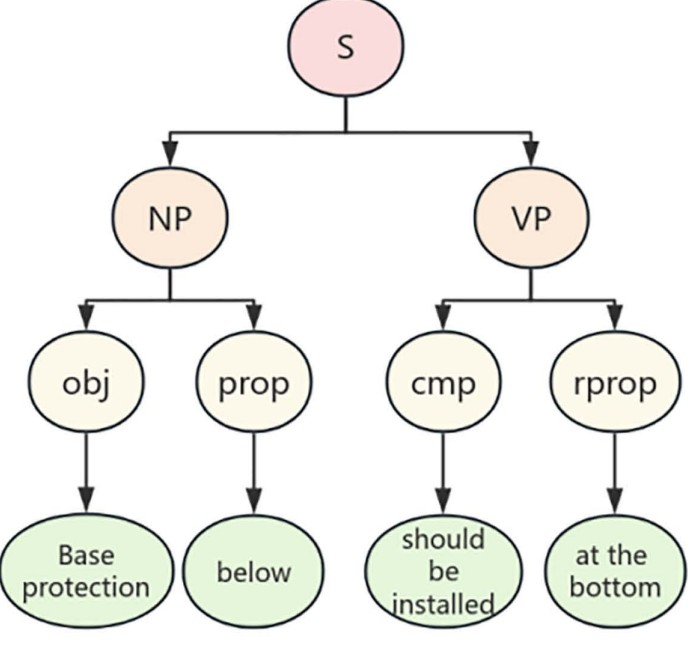

(a)Triplet tree

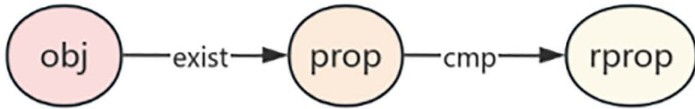

(b)Triplet template

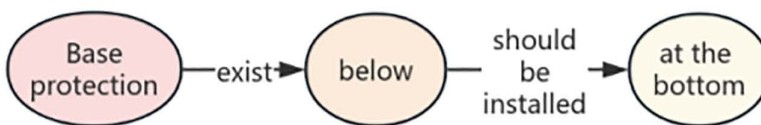

(c)Triplet conversion

**Fig 8. Relationship constraints exist.**

water surface at the entrance of the sluice. "" Length "and" 5 times "are the constraint objects," straight section of the river "can be regarded as" river "," length "is the attribute of the review object," not less than "is the comparison relationship, and the semantics are the comparison of the values of" length "and" water surface width ". Only when the conditions of" not less than "and" 5 times "are met can the specification be met. This article is expressed in a computer understandable way as a comparison between the attributes of" straight section of the river "," length "and" water

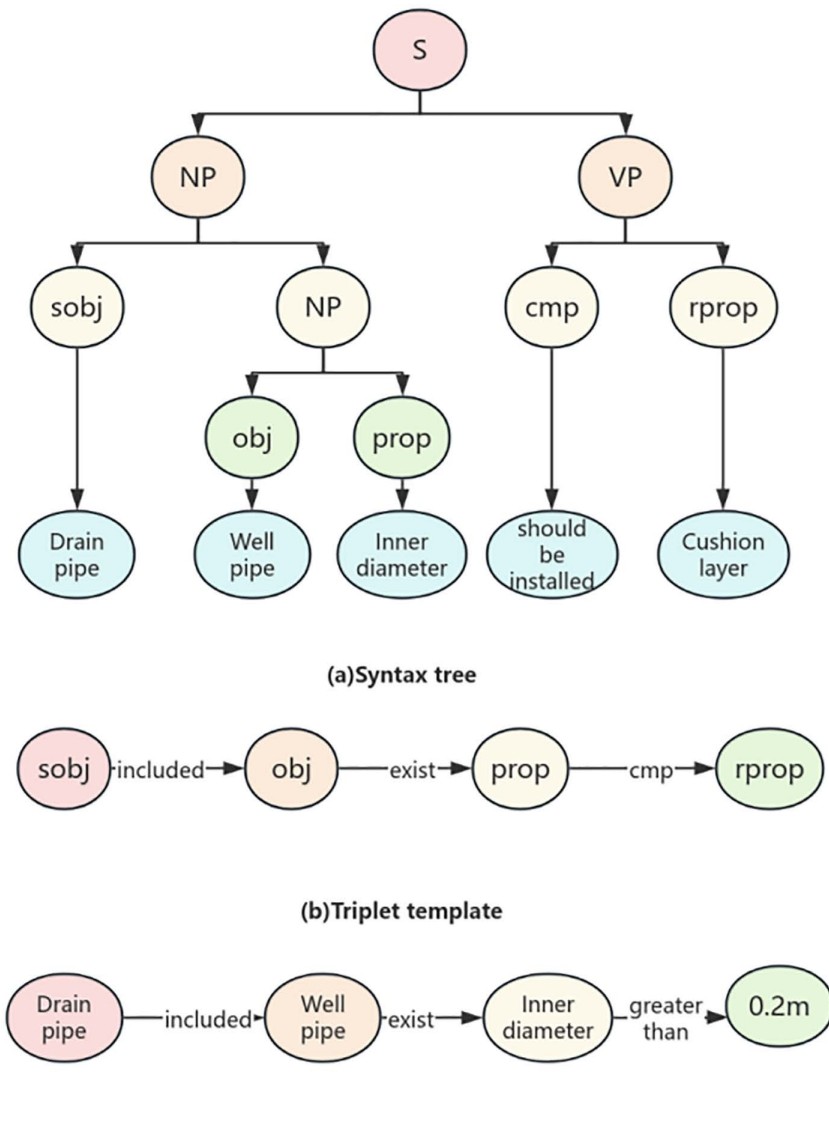

(a)Syntax tree

(b)Triplet template

(c)Triplet conversion

**Fig 9. Attribute numeric constraint.**

surface width ". When the conditions of" not less than "and" 5 times "are met, it meets the specification. The corresponding knowledge graph triplet transformation is shown in Fig 10.

(4) The nested constraint class reflects the nested structure between attributes, and examines attributes through comparative relationship constraints. Taking the example of "the bottom elevation of the traffic bridge should be 0.5m higher than the highest flood level", the "bottom elevation of the traffic bridge" includes three levels of nested "traffic bridge beam bottom elevation", where "beam" is a component of the "traffic bridge" and "bottom elevation" is an attribute of the "beam"; 'Above' is the comparative relationship, '0.5m' is the judgment condition, and both must be met simultaneously. This article can be represented by a computer as the composition of a "traffic bridge". The attribute "bottom

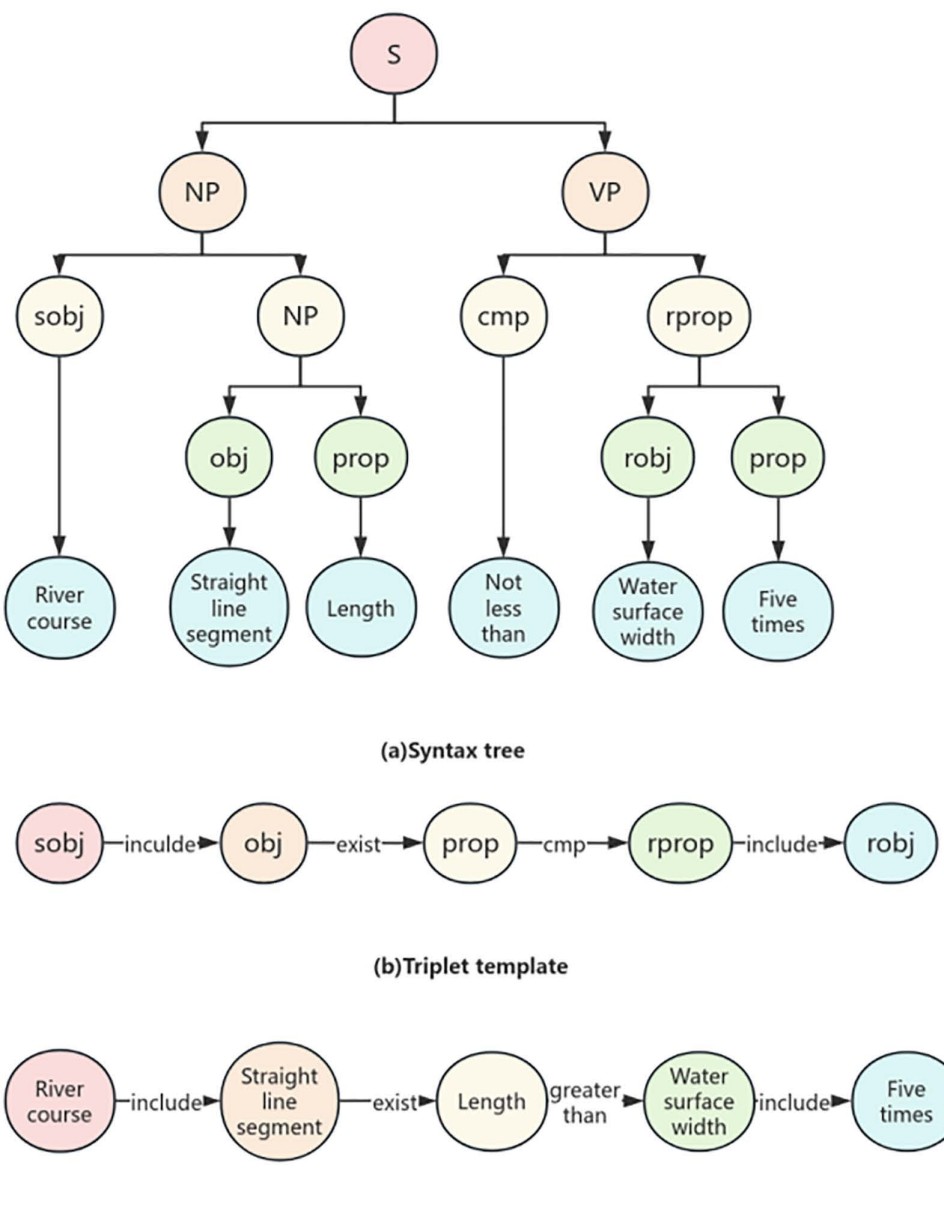

**(a)Syntax tree**

**(b)Triplet template**

**(c)Triplet conversion**

**Fig 10. Attribute scale constraint.**

elevation" of the "beam" satisfies the conditions of "0.5m higher" and "0.5m higher", which meets the specifications. The corresponding knowledge graph triplet transformation is shown in Fig 11.

(5) Conditional clauses reflect the constraint premise of subordinate clauses on the main clause through logical structure and affirmative/negative forms. Taking the example of "when the third level water gate blocks water, the safe increase value of the normal water level is not less than 0.4m", in the clause, "the third level water gate" consists of components

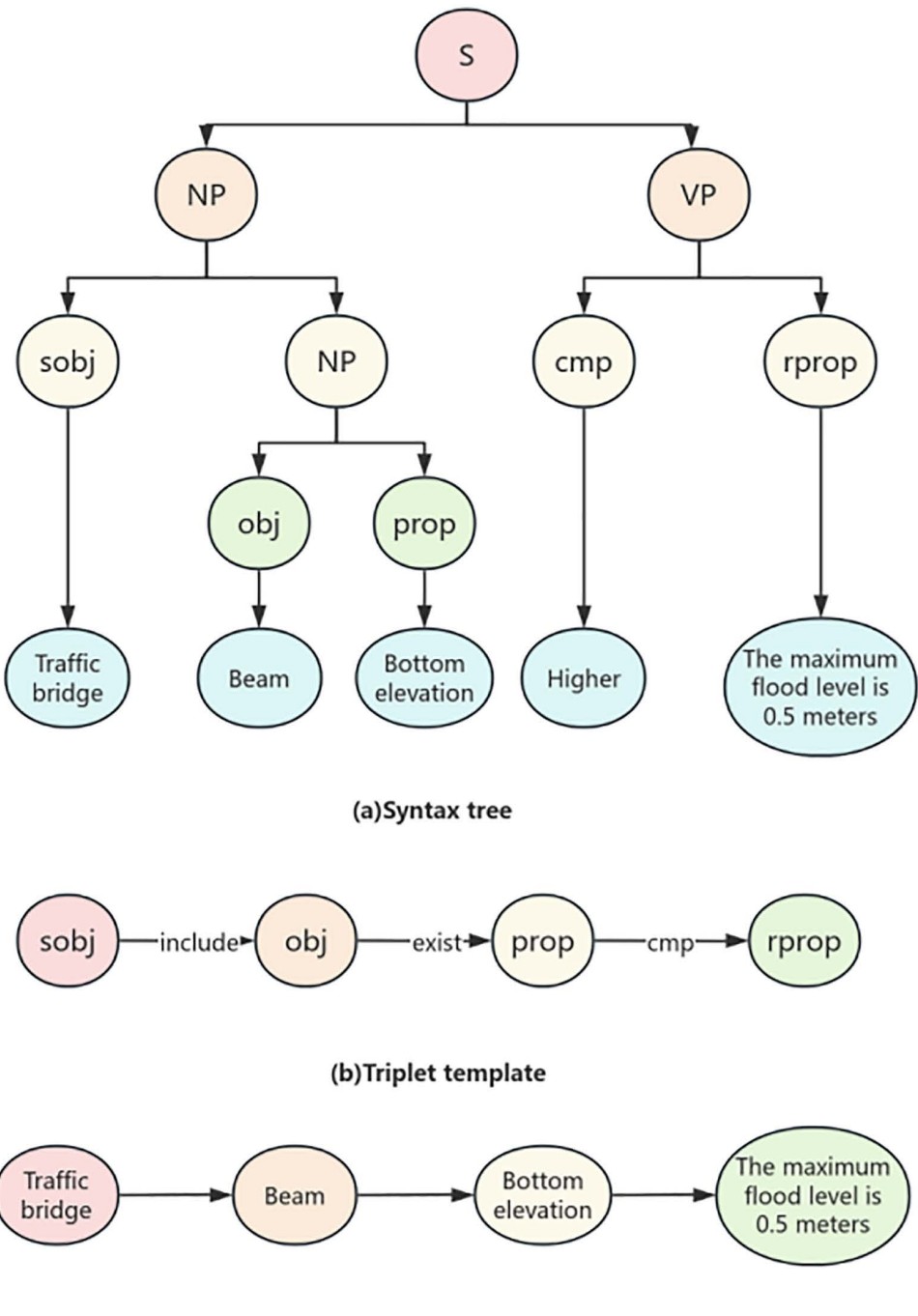

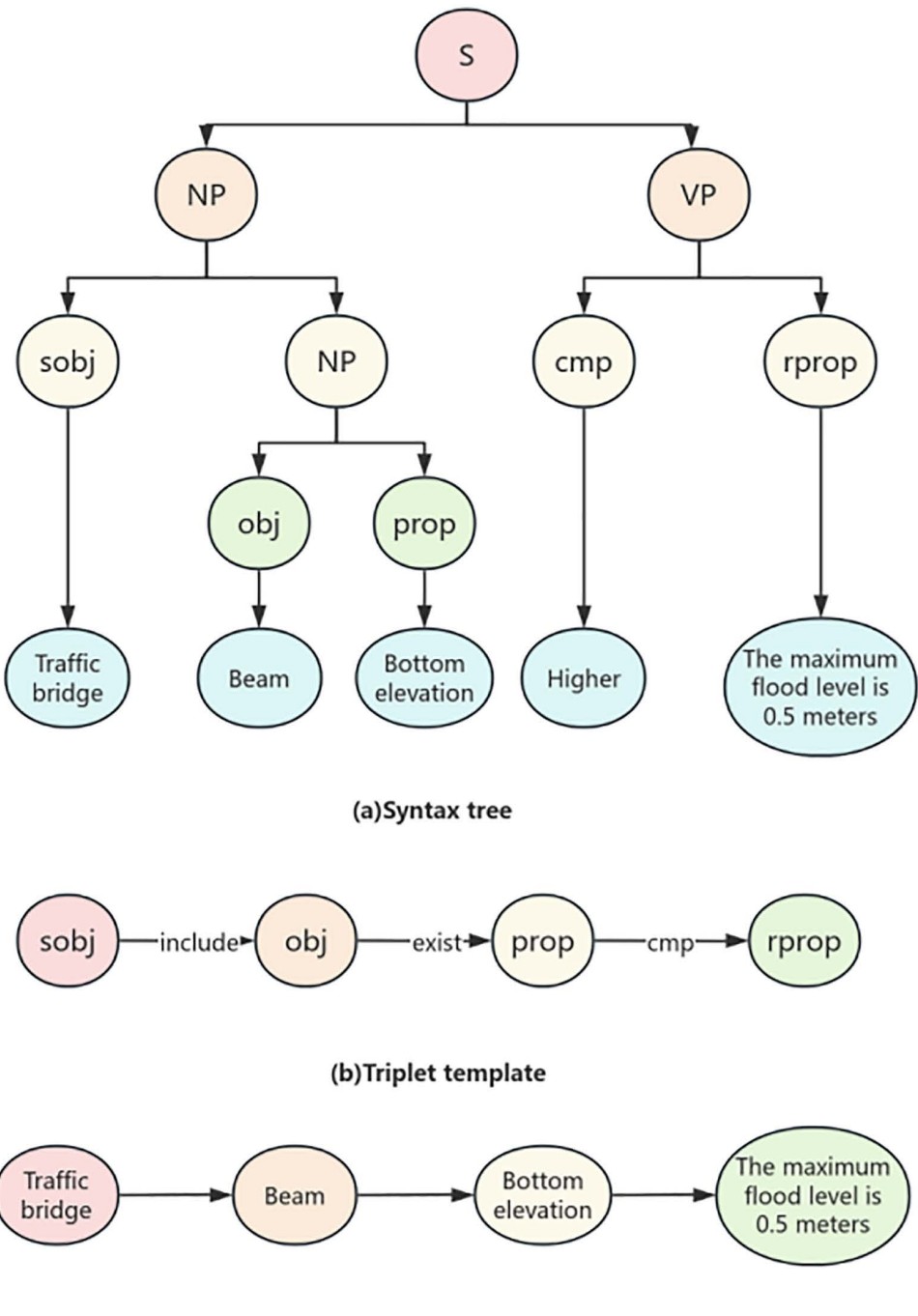

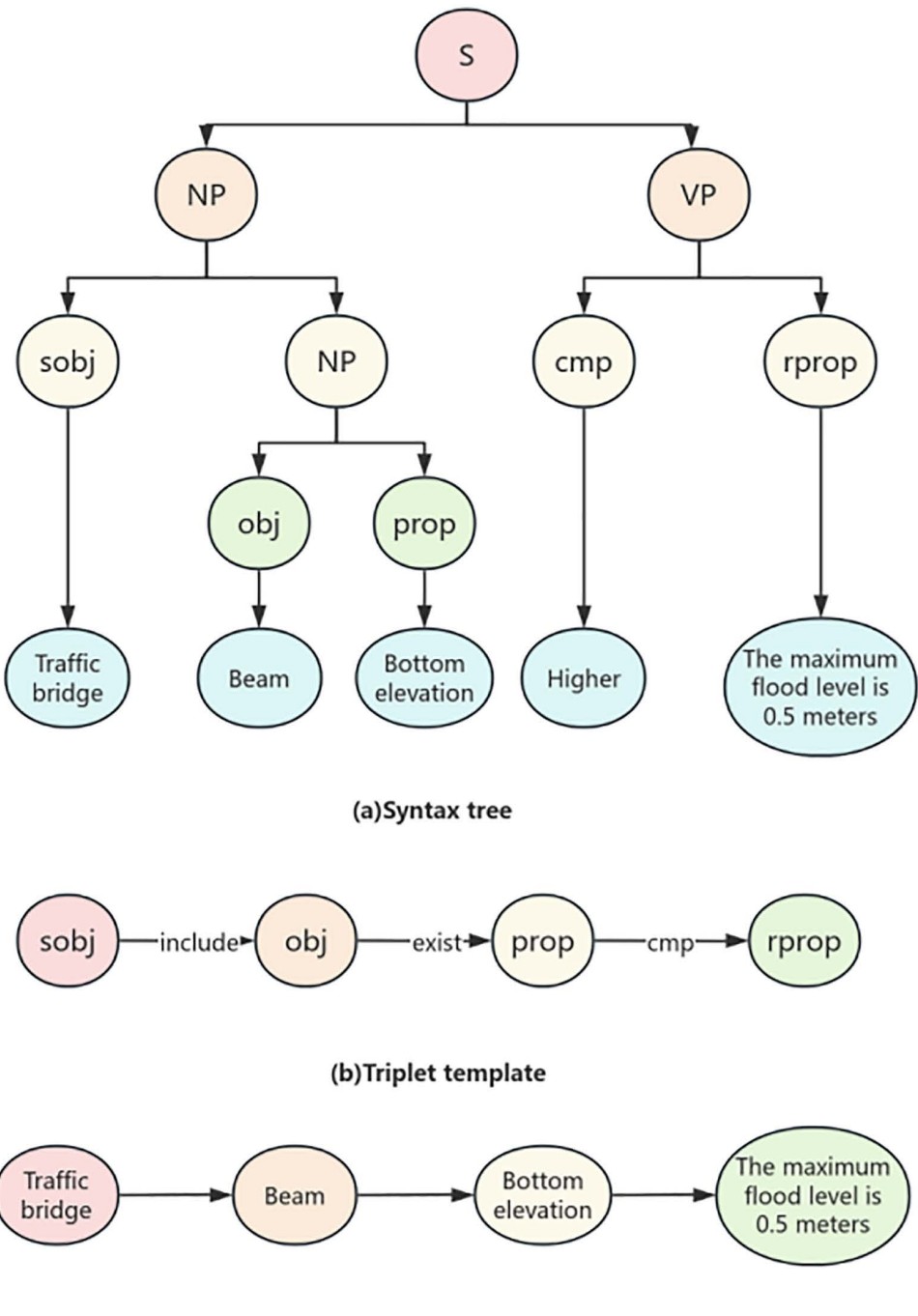

**Fig 11. Attribute nesting constraint.**

and attributes, and "blocking water" is the state condition; In the main sentence, the phrase 'safe increase value of normal water level' is composed of a noun and an attribute, while 'not lower than' is a comparative relationship. This type of provision requires first determining the conditions of the subordinate clause. When the "water gate" attribute

satisfies both "level 3" and "water blocking", it is determined whether the "safety premium value" of the "normal water level" in the main clause is "not less than" or "0.4m". If it meets the requirements, it complies with the specifications. Otherwise, the review will not pass. The knowledge graph triplet transformation is shown in Fig 12.

## Knowledge graph generation

There are three main storage solutions for knowledge graphs: relational databases, RDF triplet databases, and native graph databases. Although relational databases can store triple data ranging from tens of millions to billions, they have low efficiency in deep correlation queries and limited ability to express relationships [15]; Although RDF databases have similar semantics to graph models and good visualization effects, they have problems such as high storage overhead, insufficient design flexibility, and high query complexity [16]; Native graph databases have advantages such as high storage efficiency, convenient modeling, and strong relational expression ability, but their data capacity is relatively limited [17,18]. In response to the moderate scale of knowledge graph data in the field of water conservancy review, the need for efficient relationship expression ability, and frequent rule update requirements, this study chooses a native graph database as the storage solution [19].

The existing native graph databases include Neo4j, JanusGraph, etc. Due to Neo4j's independent storage engine, it does not rely on external systems and has good performance in graph data traversal and relational queries. Therefore, this study uses it to store knowledge of water gate specification review [20–22]. To achieve structured knowledge visualization storage and retrieval, using the Py2neo toolkit, batch process the parsed triplet statements in Cypher language and import them into Neo4j. The following is the process of importing triplet data.

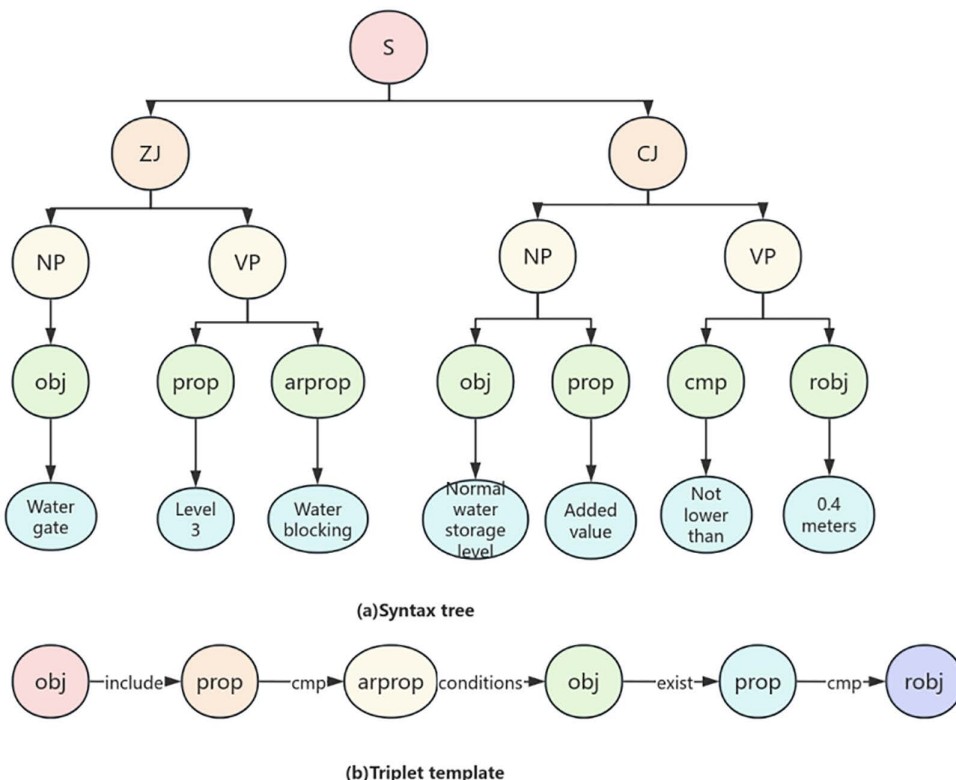

**Fig 12. Conditional constraints.**

(1) Entity import. Read standardized knowledge triplet data, add various semantic tags, and import the names of entities such as water gates, water barriers, and normal storage levels. Taking the entity nodes that import some normative knowledge as an example, as shown in Fig 13.

(2) Relationship import. Read the canonical knowledge triplet data, where each row includes a header entity, relationship, and tail entity. Add relationship labels and types, and use the CREATE statement to import all relationships. Taking the relationship between entities in the imported review articles as an example, as shown in Fig 14.

After the above steps, a preliminary knowledge graph for compliance review of water gate drawings has been constructed. Due to the large number of entities and relationships, this article presents a partial knowledge graph as shown in Fig 15.

## Conclusion

This study focuses on the in-depth research of the knowledge graph construction method for compliance review of water conservancy engineering drawings [23–25]. Through technological breakthroughs in key links such as data preprocessing and semantic annotation, the following core achievements have been formed:

(1) A systematic process for data cleaning, semantic label extraction, and structured transformation has been designed to address the characteristics of multi-source heterogeneous data in water conservancy engineering specifications. By using regular expressions to remove text noise, extracting core semantic labels based on rule matching and natural language processing techniques, and converting data into JSON structured format and syntax tree structure, a standardized data foundation is laid for knowledge graph construction.

(2) A annotation system has been constructed that includes both basic semantic tags and extended semantic tags, comprehensively covering the core review elements and complex semantic relationships of water conservancy engineering specifications. By adopting a semi-automatic annotation method, combined with automated pre annotation and manual verification, and ensuring annotation quality through cross validation, the customized annotation tool developed significantly improves annotation efficiency and accuracy.

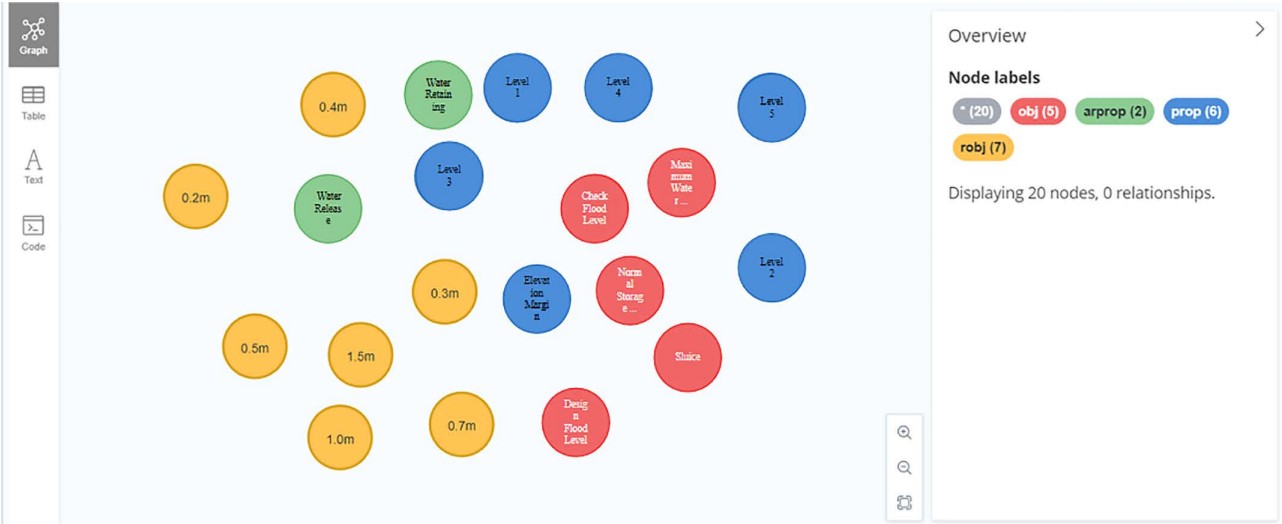

**Fig 13. Some entity import examples.**

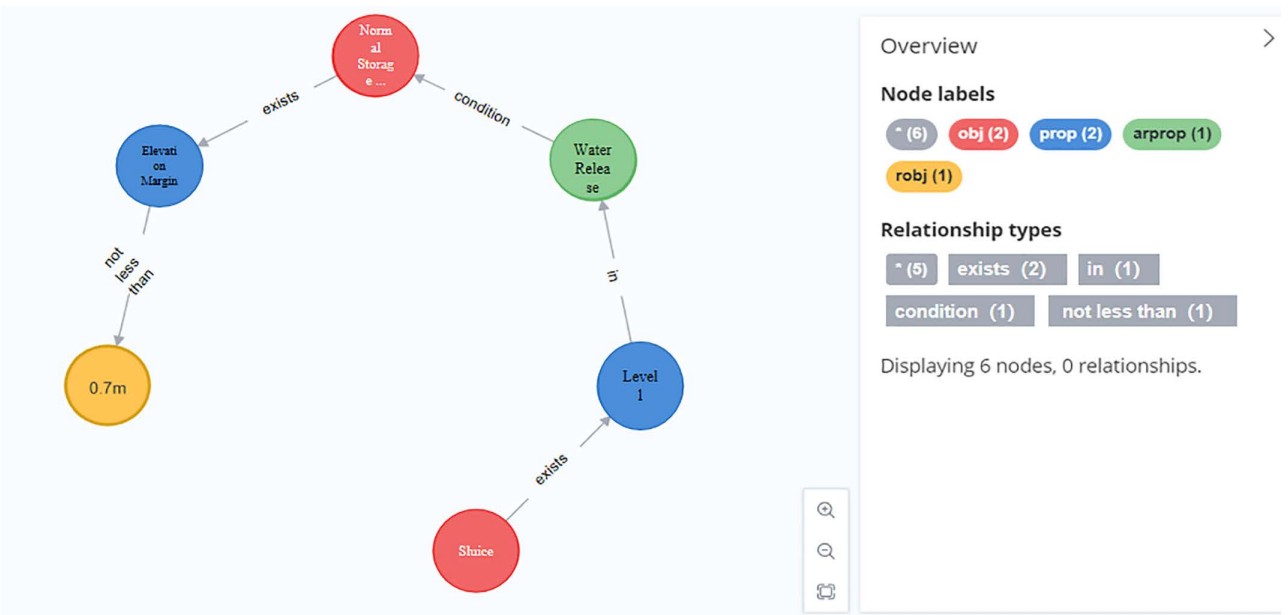

**Fig 14. Partial relation import example.**

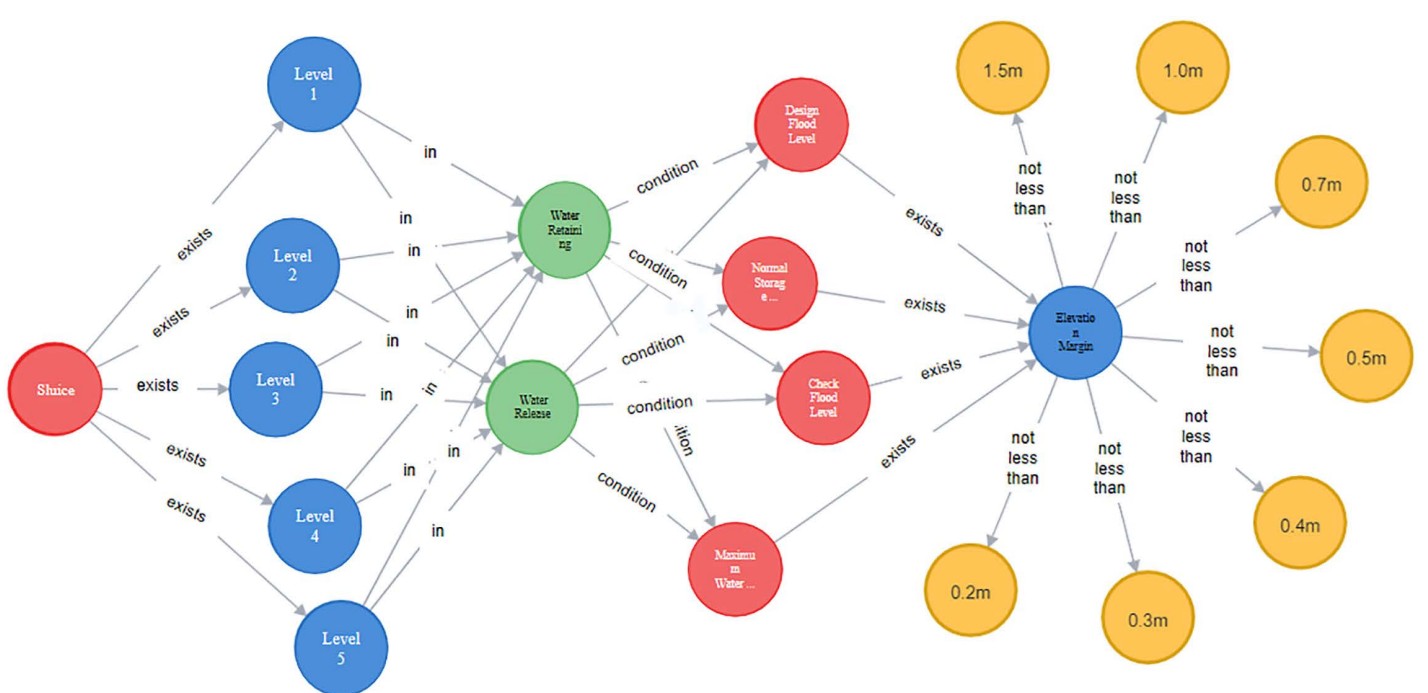

**Fig 15. Standardized knowledge map display.**

(3) The collaborative optimization of data preprocessing and semantic annotation effectively solves the problem of structuring compliance review data for water conservancy engineering reports, providing high-quality data support for entity extraction, relationship construction, and rule inference in subsequent knowledge graphs, and laying a theoretical and technical foundation for achieving automated and intelligent compliance review of water conservancy engineering reports. The knowledge graph constructed based on the optimization results can be applied to the compliance review practice of water conservancy engineering reports such as sluice drawings. Through the efficient storage and query capabilities of the Neo4j database, the review rules can be quickly retrieved and intelligently inferred, greatly improving the efficiency and accuracy of the review, and effectively promoting the development of water conservancy engineering construction towards digitalization and intelligence. It provides important technical support and practical reference for the implementation of the "smart water conservancy" strategy.

## Supporting information

**S1 File. Supporting Information.**
(DOCX)

## Author contributions

**Conceptualization:** Zelin Ding, Tao Wang, Xinhang Zhang.

**Funding acquisition:** Zelin Ding.

**Investigation:** Zhefei Fan, Xin Du.

**Methodology:** Zelin Ding, Zhefei Fan, Yuanfeng Hao.

**Project administration:** Zhefei Fan, Tao Wang.

**Resources:** Zelin Ding, Zhefei Fan.

**Software:** Zhefei Fan.

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
