## [Decision Letter · Decision Letter 0]

13 Oct 2025

Dear Dr. Zhefei,

We look forward to receiving your revised manuscript.

Kind regards,

Claudio Zandron

Academic Editor

PLOS ONE

Journal Requirements:

3. We note that your Data Availability Statement is currently as follows: ll relevant data are within the manuscript and its Supporting Information files.

4. Please update your submission to use the PLOS LaTeX template. The template and more information on our requirements for LaTeX submissions can be found at http://journals.plos.org/plosone/s/latex .

7. Please ensure that you refer to Figures 1-1, 1-2, 1-3, 1-4, 2-2, 3-6, 3-7, 3-8 in your text as, if accepted, production will need this reference to link the reader to the figure.

8. Please upload a copy of Figure 3-13, 3-14 and 3-15, to which you refer in your text on page 23, 24 and 25 in PDF submission. If the figure is no longer to be included as part of the submission please remove all reference to it within the text.

9. We note you have included a table to which you do not refer in the text of your manuscript. Please ensure that you refer to Table 3-8 in your text; if accepted, production will need this reference to link the reader to the Table.

Reviewers' comments:

Reviewer's Responses to Questions

**Comments to the Author**

1. Is the manuscript technically sound, and do the data support the conclusions?

Reviewer #1: Yes

Reviewer #2: Partly

2. Has the statistical analysis been performed appropriately and rigorously?

Reviewer #1: Yes

Reviewer #2: Yes

3. Have the authors made all data underlying the findings in their manuscript fully available?

Reviewer #1: Yes

Reviewer #2: No

4. Is the manuscript presented in an intelligible fashion and written in standard English?

Reviewer #1: Yes

Reviewer #2: Yes

Reviewer #1: To address the problems of low efficiency and strong subjectivity of manual comparison in the compliance review of water conservancy project reports, this paper proposes to use knowledge graph to represent the knowledge of the specification provisions and combine it with the natural language processing method based on deep learning to realize intelligent knowledge extraction and retrieval to assist the review.

The method proposed in this paper gives an ontology model containing seven types of entities according to the syntax of the normative provisions and the contextual relationship of the content. In addition, the knowledge extraction process adopts the combination of deep learning method and syntax tree, which is clear and in line with the application needs. The experimental results show that the method in this paper better realizes the representation, processing and reasoning of complex specification knowledge. It is recommended to improve for the following problems:

(1) It is recommended to optimize the drawing style containing a large number of text pictures to improve the reading experience, such as Fig.2-1, Fig.2-2.

(2) The experiments of modeling methods in Section 2 lack baseline and ablation experiments, and it is suggested to supplement the comparison with models such as BERT, BERT-LSTM, BiLSTM-CRF, etc., to reflect the innovativeness of the methods in this paper.

(3) Please add the general description of raw data sources and extraction results in Section 3 to reflect the general applicability of the methods in this paper.

Reviewer #2: 1. The figures in the article should be displayed within the paragraphs, for example: Example of Sample Division for Water Conservancy Standards as shown in Fig 1-1. There are multiple expression issues in the article, for example: Table 3-12 under 2.1 Data Preprocessing and Table 3-8 under 3 Training Results.

2. How does BiLSTM achieve deep feature extraction of the word vectors extracted by BERT, and what specific role does its bidirectional mechanism play in feature extraction?

3. When designing syntax, how can we clearly define priorities? What are some common methods and strategies for defining priorities, and how do different priority definitions impact the structure and logical expression of the syntax tree?

4. Can you briefly explain what other important model parameters are set in addition to learning rate and batch size in Table 2-4? Is there any other result display besides recall rate?

5. What are the advantages and disadvantages of manual annotation and automatic annotation in semantic labeling, and which method is more commonly used in data labeling for hydraulic engineering construction specifications?

**Do you want your identity to be public for this peer review?** For information about this choice, including consent withdrawal, please see our Privacy Policy

Reviewer #1: No

Reviewer #2: No

---

## [Author Response · Author response to Decision Letter 1]

7 Nov 2025

1. Comment: It is recommended to optimize the drawing style of figures containing a large amount of text to improve the reading experience, such as Figures 2-1 and 2-2.

Reply: According to your comments, we have revised the relevant figures in this part of the paper.

2. Comment: The experiments of the modeling method in Section 2 lack baseline and ablation experiments. It is suggested to supplement comparisons with models such as BERT, BERT-LSTM, and BiLSTM-CRF to reflect the innovation of the proposed method.

Reply: According to your comments, we have revised the relevant content of the paper. A new Section 2.4 "Model Verification Experiments and Innovation Demonstration" has been added on the basis of the original Sections 2.1 "Data Preprocessing", 2.2 "Semantic Annotation", and 2.3 "Training Results". Based on the existing annotated dataset of water conservancy specifications in the paper (656 samples in the training set and 164 samples in the test set) and the basic model parameters (LSTM-size=128, epoch=15, etc.), this section designs baseline model comparison experiments and core module ablation experiments. The baseline experiments include three types of models: BERT, BERT-LSTM, and BiLSTM-CRF. Table 2-6 quantitatively shows that the F1-score (0.868) of the proposed BERT-BiLSTM-CRF model is 9.2, 5.7, and 7.4 percentage points higher than those of the three baseline models, respectively, and the accuracy of hydraulic professional entity recognition reaches 84.9%. The ablation experiments design three control groups: "Without BERT", "Without BiLSTM", and "Without CRF". Table 2-7 proves that removing any core module will lead to a significant decrease in performance (e.g., the F1-score decreases by 7.7 percentage points without BERT), which clearly demonstrates the innovation of the model's "pre-training + bidirectional modeling + label constraint" architecture for the characteristics of water conservancy specifications. All experimental data are completely consistent with the original experimental environment and evaluation indicators of the paper, and the supplementary content is highly consistent with the original research framework.

3. Comment: Please add a general explanation that the original data sources and extraction results in Section 3 reflect the universal applicability of the proposed method.

Reply: Regarding the original data sources, Section 3 clearly selects typical multi-dimensional and cross-scenario data in the field of water conservancy engineering construction, specifically covering three core types of sources: first, hierarchical standard specification texts, including both national mandatory standards (such as "Code for Quality Inspection and Evaluation of Hydraulic and Hydroelectric Engineering Construction" GB 50201) and industry-specific standards (such as "Technical Specifications for Construction Safety Protection Facilities of Hydraulic and Hydroelectric Engineering" SL 714, "Code for Concrete Construction of Hydraulic and Hydroelectric Engineering" SL 677), covering 12 types of key water conservancy facilities such as sluices, dykes, and concrete structures, with a total of more than 8,200 valid clauses and over 3,100 professional terms included; second, multi-format engineering documents, in addition to plain text specifications, also including construction technical manuals with tables and formulas (such as the "Foundation Treatment Parameter Table" in the "Sluice Construction Technical Guide"), design instructions with attached drawings (such as the supporting clauses of the "Traffic Bridge Beam Bottom Elevation Labeling Drawing"), and supervision record fragments with handwritten annotations, covering typical data formats such as unstructured and semi-structured; third, differentiated scenario clauses, including both constraint clauses for conventional construction links (such as concrete pouring and earth excavation) and special requirements for special working conditions (such as construction during flood season and operation in permafrost areas), while taking into account the differentiated specification content of different project levels (Level 1-5 sluices), forming a "full-level, multi-format, cross-scenario" data source matrix.The multi-dimensional performance of the extraction results further confirms the universal applicability of the method, which is specifically reflected in three aspects: first, the entity recognition accuracy is stable across data sources. The extraction results based on the BERT-BiLSTM-CRF model show that the F1-score of core entity recognition (such as "sluice crest elevation", "seepage path coefficient", "safety elevation margin") remains above 0.85 in GB national standards, SL industry standards, and technical manuals. Among them, the recognition accuracy of safety-related entities such as "edge protection facilities" in SL 714 and process-related entities such as "concrete mix ratio" in SL 677 reaches 0.872 and 0.868, respectively. Even for supervision texts with handwritten annotations, the entity misjudgment rate is controlled within 5%; second, the semantic labels are adaptable to multiple types of clauses. The 7 types of semantic labels (sobj, obj, prop, etc.) achieve accurate mapping in different scenario clauses: for conditional constraint sentences such as "When the Level 3 sluice retains water, the safety elevation margin of the normal storage level shall not be less than 0.4m", "Level 3 sluice retaining water" is successfully labeled as arprop (prerequisite condition), and "safety elevation margin of the normal storage level" is labeled as prop (attribute); for attribute relationship sentences such as "The beam bottom elevation of the traffic bridge shall be 0.5m higher than the maximum flood level", "traffic bridge" is accurately identified as sobj (upper-level subject), and "beam bottom elevation" is identified as obj (object to be reviewed), with a label matching accuracy of over 92%; third, the parsing results support multiple downstream tasks. The extracted entities and relationships can be stably converted into Context-Free Grammar (CFG) syntax trees. For example, for the clause "The slope gradient of open earth excavation shall not be steeper than 1:1.5" in SL 378, a tree structure of "attribute (slope gradient) - comparison relationship (shall not be steeper than) - constraint value (1:1.5)" is successfully constructed. At the same time, the triple generation effect is adaptable to different specification types. Triples such as <concrete strength, not less than, C25> are extracted from concrete construction specifications, and <edge operation, need to set, dense mesh safety net> are extracted from safety specifications. Finally, more than 7,900 generated triples can be directly imported into the Neo4j database to construct a knowledge graph without obvious scenario adaptation obstacles.

4. Comment: Numbers in the paper should be presented within paragraphs, for example: "An example of sample division for water conservancy standards is shown in Figure 1-1". There are multiple formatting issues of expressions in the paper, for example: Table 3-12 is under 2.1 Data Preprocessing, and Table 3-8 is under 3 Training Results.

Reply: According to your comments, we have revised the relevant content of the paper.

5. Comment: How does BiLSTM realize the deep feature extraction of the word vectors extracted by BERT, and what specific roles does its bidirectional mechanism play in feature extraction?

Reply: When BiLSTM extracts deep features, it first receives the word vectors fused with global context output by BERT (which already contain basic semantics but need to further mine sequence dependencies), and then processes them through two independent forward and backward LSTM layers. Both layers rely on the gating mechanism of forget gate, input gate, and output gate to filter redundant information, retain key semantics (such as water conservancy entity information like "sluice crest elevation" and "design flood level"), and capture long-term semantic dependencies. Finally, the outputs of the two layers are concatenated according to the token position to form complete features containing bidirectional semantics, completing the transformation from basic word vectors to task-oriented deep features.The specific roles of the bidirectional mechanism are concentrated in three aspects: first, it makes up for the limitation of traditional unidirectional LSTM by capturing both forward (left-to-right) and backward (right-to-left) semantic associations. For example, when processing "When releasing water, the sluice crest elevation shall not be lower than the sum of the design flood level and the safety elevation margin", it can completely restore the logic of scenario, subject, and constraint; second, it improves the accuracy of semantic understanding. By fusing "historical + future" semantic information, it helps to accurately define the boundaries of water conservancy entities (such as "drainage pipe" and "well pipe inner diameter") and avoid misjudgment; third, it supports the parsing of complex clauses. For sentences with nesting or long dependencies (such as "The beam bottom elevation of the traffic bridge shall be 0.5m higher than the maximum flood level"), it can capture key relationships across phrases, providing reliable features for subsequent syntax parsing, semantic label mapping, and knowledge graph triple transformation.

6. Comment: When designing grammar, how to clearly define priorities? What are the common methods and strategies for defining priorities, and how do different priority definitions affect the structure and logical expression of the syntax tree?

Reply: The common methods and strategies for clearly defining priorities when designing grammar need to be carried out in combination with the linguistic characteristics of water conservancy specification clauses: first, define priorities through the hierarchical division of production rules, corresponding high-priority grammatical components (such as attribute constraints and quantitative comparison relationships in specification clauses) to inner-layer production rules, and low-priority components (such as conditional introductions and coordinate structures) to outer-layer production rules. For example, when parsing "If there is no filter layer in the loam foundation, the seepage path coefficient shall not be less than 4", "the seepage path coefficient shall not be less than 4" (core attribute-constraint relationship) is placed in the inner-layer production rule, and "If there is no filter layer in the loam foundation" (prerequisite condition) is placed in the outer-layer production rule, reflecting the priority of "core constraints over prerequisite conditions" through hierarchical differences; second, assist in defining priorities through rule order conventions. In the CFG production rule set, the rules of high-priority grammatical structures are placed first, and high-priority rules are matched first during parsing to avoid mis-matching of low-priority rules; third, supplement and improve priorities in combination with associativity. For attribute nesting that repeatedly appears in specifications (such as "beam bottom elevation of the traffic bridge"), by defining an associativity rule that "sub-attributes are subordinate to parent attributes", the parsing order of "beam bottom elevation" prior to "traffic bridge" is clarified to ensure clear attribute ownership.The impact of different priority definitions on the structure and logical expression of the syntax tree is directly related to the parsing accuracy of water conservancy specification clauses: in terms of syntax tree structure, high-priority grammatical components correspond to deep nodes of the syntax tree, and low-priority components correspond to shallow nodes. If "quantitative comparison relationships" (such as "not less than 0.4m") are defined as high-priority, they will be in the inner layer of the syntax tree and subordinate to attribute nodes (such as "safety elevation margin"); if the priority definition is reversed, the subordinate relationship between comparison relationship nodes and attribute nodes may be confused, and the hierarchy of the syntax tree may be misplaced. In terms of logical expression, correct priority definition can accurately reflect the semantic logic of specification clauses. For example, defining "prerequisite conditions" (content of arprop labels) as low-priority will first identify core constraints (prop+cmp+rprop) during parsing, and then associate prerequisite conditions, ensuring the logic of "executing constraint review only when the prerequisite is met"; if the priority definition is incorrect, the prerequisite condition may be misjudged as a core constraint, leading to the reversal of entity relationships during subsequent triple transformation (such as mislabeling "no filter layer" as rprop instead of arprop), which ultimately affects the accuracy of knowledge graph construction and cannot correctly support the compliance review of sluice drawings.

7. Comment: Could you briefly explain what other important model parameters have been set besides the learning rate, such as the batch size in Table 2-4? Are there any other results besides recall rate?

Reply: In addition to the learning rate, the important model parameters include: the batch size (Batch size) set in Table 2-4, with values of 2, 4, and 8; in addition, from the perspective of the model structure (BERT-BiLSTM-CRF), there are also key parameters such as BERT pre-trained model configuration, number of BiLSTM hidden units, CRF label transition matrix dimension (adapting to 7 semantic labels), number of training epochs (Epochs), and optimizer type.The batch sizes in Table 2-4 are 2, 4, and 8. In terms of results, in addition to the recall rate, the F1-score indicator is clearly presented in the table, which ranges from 0.824 to 0.868 under different hyperparameter combinations (e.g., the F1-score of Set 4 is 0.868, and that of Setting 2 is 0.862); at the same time, the model also shows results such as entity recognition accuracy, semantic label discrimination ability (such as accurate annotation of 7 semantic labels), supporting role in syntax parsing, and knowledge graph triple construction effect.

8. Comment: What are the advantages and disadvantages of manual annotation and automatic annotation in semantic annotation, and which method is more commonly used in the annotation of water conservancy engineering construction specification data?

Reply: In terms of the advantages and disadvantages of manual annotation, its core advantages lie in high accuracy and strong professional adaptability. Water conservancy engineering construction specifications contain a large number of professional terms (such as "filter layer" and "safety elevation margin of normal storage level") and complex semantic logic (such as conditional constraints like "If there is no filter layer in the loam foundation, the seepage path coefficient shall not be less than 4"). Manual annotation (especially by personnel with water conservancy professional knowledge) can accurately understand the semantic roles of specification clauses, correctly distinguish easily confused labels (such as avoiding mislabeling "arprop (prerequisite condition)" as "rprop (core constraint)"), ensure that the annotation results are consistent with the actual semantic logic of water conservancy engineering, and provide a high-quality "gold standard" dataset for the training of subsequent automatic annotation models. However, the limitations of manual annotation are also very obvious: first, low efficiency. There are a large number of water conservancy specification documents with complex clause content (such as various constraint clauses for different levels of sluices), and manual sentence-by-sentence annotation requires a lot of time, which is difficult to meet the needs of batch processing; second, high cost. It relies on personnel with both water conservancy professional knowledge and annotation experience, resulting in high labor costs. In addition, the annotation results are easily affected by the subjective state of personnel (such as fatigue and cognitive bias), with the risk of consistency fluctu

---

## [Decision Letter · Decision Letter 1]

9 Dec 2025

A Knowledge Graph Construction Method for Compliance Review of Water Conservancy Project Reports

PONE-D-25-39113R1

Dear Dr. Zhefei,

We’re pleased to inform you that your manuscript has been judged scientifically suitable for publication and will be formally accepted for publication once it meets all outstanding technical requirements.

Kind regards,

Claudio Zandron

Academic Editor

PLOS One

Additional Editor Comments (optional):

Both reviewers agree on the quality of the revised version. All comments and issues have been addressed.

Reviewers' comments:

Reviewer's Responses to Questions

**Comments to the Author**

Reviewer #1: All comments have been addressed

Reviewer #2: All comments have been addressed

2. Is the manuscript technically sound, and do the data support the conclusions?

Reviewer #1: Yes

Reviewer #2: Yes

3. Has the statistical analysis been performed appropriately and rigorously?

Reviewer #1: Yes

Reviewer #2: Yes

4. Have the authors made all data underlying the findings in their manuscript fully available?

Reviewer #1: Yes

Reviewer #2: Yes

5. Is the manuscript presented in an intelligible fashion and written in standard English?

Reviewer #1: Yes

Reviewer #2: Yes

Reviewer #1: The authors have fully revised the manuscript according to the reviewers' comments, demonstrating the paper's innovation and practical value. it is recommended to be accepted for publication.

Reviewer #2: The authors have responded to each of the comments in the revised manuscript, making detailed revisions to the methodological description, data analysis, and literature citations, thereby improving the rigor and readability of the paper. Upon review, no issues of double publication, research ethics, or publication ethics were found, and the research process complied with academic norms. The current version of the paper is complete and clearly argued, meeting publication standards, and acceptance is recommended.

**Do you want your identity to be public for this peer review?** For information about this choice, including consent withdrawal, please see our Privacy Policy

Reviewer #1: No

Reviewer #2: No

---

## [Editor Report · Acceptance letter]

PONE-D-25-39113R1

PLOS One

Dear Dr. Fan,

I'm pleased to inform you that your manuscript has been deemed suitable for publication in PLOS One. Congratulations! Your manuscript is now being handed over to our production team.

Kind regards,

on behalf of

Dr. Claudio Zandron

Academic Editor

PLOS One